# Brown Tumour in Chronic Kidney Disease: Revisiting an Old Disease with a New Perspective

**DOI:** 10.3390/cancers15164107

**Published:** 2023-08-15

**Authors:** Djoko Santoso, Mochammad Thaha, Maulana A. Empitu, Ika Nindya Kadariswantiningsih, Satriyo Dwi Suryantoro, Mutiara Rizki Haryati, Decsa Medika Hertanto, Dana Pramudya, Siprianus Ugroseno Yudho Bintoro, Nasronudin Nasronudin, Mochamad Yusuf Alsagaff, Hendri Susilo, Citrawati Dyah Kencono Wungu, Nicolaas C. Budhiparama, Pancras C. W. Hogendoorn

**Affiliations:** 1Department of Internal Medicine, Dr. Soetomo Hospital, Surabaya 60286, Indonesia; djoko-santoso@fk.unair.ac.id (D.S.); decsa_medika@yahoo.com (D.M.H.); danapramudya@gmail.com (D.P.); ugroseno@fk.unair.ac.id (S.U.Y.B.); nasronudindr@yahoo.com (N.N.); 2Department of Internal Medicine, Faculty of Medicine, Universitas Airlangga Hospital, Universitas Airlangga, Surabaya 60115, Indonesia; satriyo.dwi.suryantoro@fk.unair.ac.id (S.D.S.); drmutiara.rh@gmail.com (M.R.H.); 3Department of Anatomy, Histology, and Pharmacology, Faculty of Medicine, Universitas Airlangga, Surabaya 60132, Indonesia; maulana.antiyan@fk.unair.ac.id; 4Department of Microbiology, Faculty of Medicine, Universitas Airlangga, Surabaya 60132, Indonesia; ika.nindya@fk.unair.ac.id; 5Department of Cardiology and Vascular Medicine, Universitas Airlangga Hospital, Universitas Airlangga, Surabaya 60115, Indonesia; yusuf_505@fk.unair.ac.id (M.Y.A.); hendrisusilo@staf.unair.ac.id (H.S.); 6Department of Physiology and Medical Biochemistry, Faculty of Medicine, Universitas Airlangga, Surabaya 60132, Indonesia; citrawati.dyah@fk.unair.ac.id; 7Department of Orthopaedic Surgery, Leiden University Medical Center, Albinusdreef 2, 2333 ZA Leiden, The Netherlands; n.c.budhiparama@lumc.nl; 8Department of Pathology, Leiden University Medical Center, Albinusdreef 2, 2333 ZA Leiden, The Netherlands

**Keywords:** bone tumour, hyperparathyroidism, giant cell, osteoclast, bone neoplasm

## Abstract

**Simple Summary:**

OFC (Osteitis Fibrosa Cystica) and Brown Tumours, skeletal lesions commonly found in chronic kidney disease (CKD) patients, are influenced by various risk factors, such as age, sex, medications affecting calcium metabolism, and vitamin D deficiency. The primary cause is secondary hyperparathyroidism, leading to imbalances in calcium and phosphorus levels and osteoclast activation. Other factors, like RAAS hyperactivity and chronic inflammation, may also contribute to their development. The recently described involvement of KRAS mutations turned Brown Tumours from reactive lesions to potentially neoplastic lesions. To manage these conditions, pharmacologic treatments like bisphosphonates, calcimimetics, vitamin D supplementation, and denosumab can help by reducing hyperparathyroidism, restoring calcium levels, and preventing OFC occurrence. Brown Tumours, being rare, lack sufficient understanding regarding their manifestation and treatment. However, considering their impact on CKD patients’ quality of life, it is crucial for nephrologists and medical practitioners working with dialysis patients to be aware of various diagnostic and treatment options.

**Abstract:**

Osteitis fibrosa cystica (OFC) and Brown Tumours are two related but distinct types of bone lesions that result from the overactivity of osteoclasts and are most often associated with chronic kidney disease (CKD). Despite their potential consequences, these conditions are poorly understood because of their rare prevalence and variability in their clinical manifestation. Canonically, OFC and Brown Tumours are caused by secondary hyperparathyroidism in CKD. Recent literature showed that multiple factors, such as hyperactivation of the renin–angiotensin–aldosterone system and chronic inflammation, may also contribute to the occurrence of these diseases through osteoclast activation. Moreover, hotspot KRAS mutations were identified in these lesions, placing them in the spectrum of RAS–MAPK-driven neoplasms, which were until recently thought to be reactive lesions. Some risk factors contributed to the occurrence of OFC and Brown Tumours, such as age, gender, comorbidities, and certain medications. The diagnosis of OFC and Brown Tumours includes clinical symptoms involving chronic bone pain and laboratory findings of hyperparathyroidism. In radiological imaging, the X-ray and Computed tomography (CT) scan could show lytic or multi-lobular cystic alterations. Histologically, both lesions are characterized by clustered osteoclasts in a fibrotic hemorrhagic background. Based on the latest understanding of the mechanism of OFC, this review elaborates on the manifestation, diagnosis, and available therapies that can be leveraged to prevent the occurrence of OFC and Brown Tumours.

## 1. Introduction

More than 70% of patients undergoing dialysis are affected by renal osteodystrophy [1]. Osteitis fibrosa cystica (OFC), also known as von Recklinghausen’s disease of the bone and Brown Tumour, are two related but distinct types of bone lesions that are associated with chronic kidney disease and hyperparathyroidism [2]. OFC is characterized by excessive osteoclast activation/production, leading to the resorption of bone and cyst formation [2,3]. OFC was first described by Engel in 1864 and von Recklinghausen in 1891, though the first recorded case was in the collection of Hunter, who lived from 1718 to 1783 [4,5]. Brown Tumour is a more aggressive tumour-forming lesion of OFC and is characterized by the presence of brown pigment in the lesion [6,7]. The term was first coined for a tumour-like lesion by Henry Jaffe [8]. Brown Tumours can present as a solitary lesion, but multiple lesions within one patient are not uncommon (Figure 1 and Figure 2). The pathophysiology of these conditions is not well understood.

There have been several previous chronic kidney disease (CKD) cases in which Brown Tumours can be found [6,9,10]. CKD is a common condition that is characterized by a progressive loss of kidney function for more than 3 months and affects 8% to 16% of the population worldwide [11,12]. Haemodialysis is a common treatment for patients with advanced CKD. These patients are at increased risk of developing OFC and Brown Tumours [2,13].

Primary hyperparathyroidism is another important factor in the development of OFC and Brown Tumours [14,15,16]. It is a condition in which the (hyperplastic) parathyroid glands or adenoma of the parathyroid produce excessive amounts of parathyroid hormone (PTH), leading to an imbalance in calcium and phosphorus metabolism [17]. This can result in the hyperactivation of osteoclasts and the subsequent destruction of bone tissue, leading to the development of OFC and Brown Tumours [17,18].

The exact mechanisms by which chronic kidney disease and hyperparathyroidism lead to the development of OFC and Brown Tumours are not fully understood. As previously stated, PTH production and other factors, such as cytokines and growth factors, may play a role in the pathogenesis of these conditions [17,18]. Numerous mediators such as NF-κB, receptor activator of NF-κB ligand (RANKL), and macrophage colony-stimulating factor (M-CSF) were involved in the downstream signalling of PTH and calcium homeostasis [17,18].

Here, we aim to provide a comprehensive overview of the current understanding of OFC and Brown Tumours in CKD. We start by discussing the epidemiology of these conditions in patients with CKD. We will then review the current understanding of the mechanisms by which CKD and hyperparathyroidism can lead to the development of OFC and Brown Tumours. Some factors, such as hyperparathyroidism, hyperstimulation of the renin–angiotensin–aldosterone system (RAAS), and chronic inflammation, are discussed in relation to osteoclast activation. Finally, we will discuss the potential treatment options for these conditions and the challenges and gaps in current knowledge.

## 2. Epidemiology and Risk Factors

The exact prevalence of OFC and Brown Tumours is not well known, but they are more commonly found in patients with advanced CKD who are on hemodialysis [13]. In a study by Jat et al. (2016), it was stated that the prevalence of OFC in end-stage renal disease was 32% [19]. In contrast, Brown Tumours are relatively rare, with a prevalence of about 2% among patients with secondary hyperparathyroidism [20].

There are several risk factors associated with an increased risk of developing these conditions, including age, gender, comorbidities, and certain medications. OFC and Brown Tumours are more common in older patients, with a peak incidence in the sixth and seventh decades of life [21,22,23]. This may be due to an age-related decline in renal function, which can affect the kidneys’ ability to regulate calcium and phosphorus metabolism.

Gender is also a significant risk factor. Specifically, females are more commonly affected, with a male-to-female ratio of 1:3 in ages below 30 [20]. The exact reasons for this gender difference are not well understood, but hormonal and genetic factors have been suggested to play a role. The present comorbidities would influence the quality of life of patients with OFC and Brown Tumours. Spontaneous tendon ruptures, pruritus from calcium deposits in the skin, ocular calcification, and calcification of the joints also frequently accompany secondary hyperparathyroidism in CKD [24,25]. Certain medications can also increase the risk of developing these conditions. Calcium supplementation, vitamin D deficiency, and the use of glucocorticoids have all been associated with an increased risk of OFC and Brown Tumours in patients with CKD [24,25].

In summary, age, sex, and certain medications are all significant risk factors for the development of OFC or Brown Tumours in patients with CKD; however, the exact mechanisms of how they influence the occurrence and progression of the disease are as yet unknown.

## 3. The Mechanisms: Beyond Hyperparathyroidism

The exact mechanisms by which CKD and hyperparathyroidism lead to the development of OFC and Brown Tumours are not fully understood. Some studies have suggested that PTH production and other factors, such as RAAS activation, cytokine production, and increased growth factor expression, may play a role in the pathogenesis of these conditions [17,18].

### 3.1. Secondary Hyperparathyroidism

Secondary hyperparathyroidism, which classically occurs during CKD, is central to the formation of Brown Tumours [9,26]. The parathyroid glands are responsible for regulating calcium ion homeostasis by modulating bone metabolism, the synthesis of 1α, 25-dihydroxy vitamin D (1α,25(OH)2D) in proximal tubules, and the reabsorption of the calcium ion [17,27].

There are several factors involved in the pathogenesis of secondary HPT in CKD. Phosphate retention, hyperphosphatemia, low serum calcium ion, deficiency of 1α,25(OH)2D deficiency, elevated levels of PTH, intestinal calcium malabsorption, and the reduction of vitamin D receptors (VDRs) and calcium-sensing receptors (CaSRs) in the parathyroid glands play a role in the development of secondary hyperparathyroidism [17,28,29,30,31,32,33]. Furthermore, parathyroid hyperplasia is often present. Based on these observations concerning the pathogenesis, therapy for secondary HPT in the context of CKD and ESRD includes controlling serum phosphate concentrations, administering calcium and vitamin D analogs, and administering calcimimetics [34,35,36,37].

In hyperparathyroidism, excessive production of PTH leads to an imbalance in calcium and phosphorus metabolism [14]. This can result in the activation of osteoclasts, which are cells that are responsible for the breakdown and removal of bone tissue [21]. Osteoclasts do not express functional PTH receptors. Therefore, PTH-induced increases in osteoclast activity and number result from non-cell-autonomous pathways. The two main cytokines that promote osteoclast differentiation and function are M-CSF and RANKL (Figure 3) [21]. Furthermore, M-CSF is secreted and binds to its receptor, c-Fms, on the surface of osteoclast precursor cells [38,39]. The binding of M-CSF to c-Fms triggers a series of intracellular signalling events that promote the survival, proliferation, and differentiation of osteoclast precursor cells into mature osteoclasts. It also enhances the expression of other crucial factors necessary for osteoclast function, such as RANK (Receptor Activator of Nuclear Factor Kappa-B) and NFATc1 (Nuclear Factor of Activated T cells, cytoplasmic 1) [38,39,40].

It has been established that PTH increases the expression of M-CSF and RANKL [21]. The bone microenvironment has numerous biological sources of both of these cytokines, including marrow stromal cells, osteoblasts, resident marrow lymphocytes, and osteocytes. Numerous cell types that express PTH receptors contain the well-researched PTH target gene RANKL [21]. The binding of expressed RANKL to the RANK receptor on the surface of monocyte/macrophage precursor cells triggers a series of intracellular signalling pathways, including the NF-κB and MAPK pathways, ultimately leading to the activation of transcription factors like NFATc1 [39]. This activation regulates osteoclastogenesis and induces the expression of genes required for osteoclast differentiation, such as tartrate-resistant acid phosphatase (TRAP) and cathepsin K, which are essential for bone resorption. As a result of RANKL stimulation and NFATc1 activation, monocyte/macrophage precursor cells undergo fusion, forming large, multinucleated cells called osteoclasts. These multinucleated osteoclasts (Figure 4) attach to the bone surface and secrete enzymes, including acids and proteases, which dissolve the mineral and organic matrix of the bone, resulting in bone resorption [38,39]. Alternatively, osteocyte-derived RANKL is necessary for secondary hyperparathyroidism-induced increases in osteoclasts and bone loss. In OFC, excessive activation of osteoclasts leads to the destruction of bone tissue and the formation of cystic lesions [21].

Through previous studies, serum alkaline phosphatase (ALP) concentrations of OFC and Brown Tumours tended to elevate over the normal range [41,42,43,44]. ALP is highly expressed in the cells of mineralized tissue, such as bone and liver, and plays a critical function in tissue formation by reducing the extracellular pyrophosphate concentration, an inhibitor of mineral formation [45]. The elevated ALP group correlated with significantly higher levels of preoperative PTH and phosphorus, lower serum calcium, and a longer hospital stay for secondary hyperparathyroidism in CKD [46]. Preoperative PTH and ALP levels can predict postoperative calcium requirements [47,48]. Therefore, the existing status of this critical enzyme should be reviewed periodically [45].

In Brown Tumours, the calcium imbalance induces vascular calcification and fragility. The excessive vascularization and focal haemorrhage become the hallmarks of the lesion, which differentiates them next to their size from OFC. The breakdown of red blood cells caused the accumulation of hemosiderin, which appears as a brown pigment [23,49].

### 3.2. RAAS Hyperactivity

Other than secondary hyperparathyroidism, individuals with CKD are at high risk of osteoclast activation (Figure 3). One of the leading comorbidities and cardiovascular risk factors in CKD is hypertension due to renin–angiotensin–aldosterone system (RAAS) activation [50,51]. Considering that angiotensin II can induce the expression of RANKL in osteoblasts, it could lead to the activation of osteoclasts [52].

Multinucleated cells of the monocyte/macrophage lineage, which can also be angiotensin II targets, are the source of osteoclasts [53]. This is because they have a unique ability to undergo the specialized process known as osteoclastogenesis. Osteoclasts are large, multinucleated cells responsible for bone resorption, which is essential for maintaining bone health, remodelling, and repair [54]. The process of osteoclastogenesis begins with the differentiation of hematopoietic stem cells into monocytes [55]. Monocytes are a type of white blood cell and part of the mononuclear phagocyte system, which also includes macrophages. These cells are derived from the bone marrow and circulate in the bloodstream. Under specific physiological conditions, such as bone remodelling in response to mechanical stress or injury, certain signalling molecules and cytokines are produced in the bone microenvironment [54,55]. To direct the differentiation of osteoclasts from their precursors, osteoblasts and stromal cells express RANKL in response to many bone-resorbing agents, including vitamin D3 [56]. Through cell-to-cell contacts with osteoblasts and stromal cells, osteoclast precursors identify RANKL and develop into mature osteoclasts [52]. Thus, targeted disruption of RANKL results in an osteoporotic phenotype and a lack of osteoclasts. Clinical studies have shown that angiotensin-converting enzyme (ACE) inhibitors decrease the risk of fractures and improve bone metabolism [57,58].

### 3.3. Inflammatory Factors Induce Osteoclast Activation

Patients with CKD are at risk of developing chronic inflammation caused by infection, uraemic milieu, or tissue ischemia [59,60]. Intrinsic damage-associated molecular patterns (DAMPs), which are generated by cells after cell death or tissue remodelling, frequently cause this inflammation [59,60]. Additionally, various parenchymal cell types express Toll-like receptors (TLRs) and inflammasome components. Numerous pro-inflammatory cytokines such as interleukin (IL)-1β, IL-6, and Tumour necrosis factor (TNF)-α are consistently increasing across different stages of CKD [60,61]. These trigger vascular dysfunction and the innate immune response, which cause microinflammation [59,60].

Inflammatory factors may also play a role in the development of OFC and Brown Tumours. Immune cell-produced cytokines and growth factors can contribute to excessive osteoclast activation and subsequent destruction of bone tissue (Figure 3). A previous study reported that IL-1β activated osteoclasts in the presence of M-CSF and RANKL [59]. Osteoclasts in healthy bones express both type 1 and type 2 IL-1 receptors at similar levels. However, pathologically activated osteoclasts preferentially express the stimulatory IL-1 receptor type 1, which is prone to induce osteoclast activation [59,62].

### 3.4. Molecular Drivers of Tumour Formation

Recently, it became clear that in the majority of Brown Tumours, hotspot somatic KRAS mutations are present [63]. This sheds new light on these tumours, which were tilled and then considered reactive instead of neoplastic lesions. As a hypothesis, the authors suggest that the regression of Brown Tumours after normalization of hyperparathyroidism is the result of a second genetic hit mediated by endocrine stimulation [63].

This mutation in Brown Tumours underpins its relation to non-ossifying fibroma and giant cell granuloma of the jaws, which are also driven by RAS/MAPK signalling [64]. In RAS/MAPK signalling (Figure 5A), the recruited adapter proteins activate RAS proteins by promoting the exchange of GDP (guanosine diphosphate) for GTP (guanosine triphosphate) on the RAS proteins [65]. Then, the RAS proteins initiate a kinase cascade, starting with the activation of RAF (Rapidly Accelerated Fibrosarcoma) kinases. RAF kinases then phosphorylate and activate MEK (MAPK/ERK Kinase) proteins [65]. The activated MEK proteins, in turn, phosphorylate and activate MAPKs, including ERK1 (Extracellular Signal-Regulated Kinase 1) and ERK2 [66]. These activated MAPKs then translocate to the cell nucleus. In the nucleus, ERKs phosphorylate various transcription factors and co-regulators, altering gene expression and influencing cellular responses. This can lead to the activation or repression of specific genes involved in cell proliferation, survival, and differentiation. The RAS/MAPK pathway is tightly regulated to maintain cellular homeostasis, as dysregulation of this pathway can lead to various diseases, including cancer [67,68]. Mutations or overactivation of RAS proteins, including KRAS mutations in certain cancers, can drive uncontrolled cell growth and contribute to tumour formation [64,67,68].

It may also explain the morphological similarity between the lesions. Interestingly, in non-ossifying fibromas, only a subset of the cells carry the mutation. Likewise, in Brown tumours, the interplay between reactive and neoplastic cells may play a role in sustaining growth, and it might be speculated that OFC is the non-neoplastic precursor lesion that can turn into a Brown Tumour following the induction of the KRAS mutation [69,70].

Similar to Giant Cell Lesions, mutations in KRAS have been identified in Brown Tumours, which lead to MAPK/ERK activation [69]. The most prevalent KRAS activating mutations found were related to p.A146P/T/V, p.G12D/V, p.G13C/D, p.K117R, and p.Q22K, affecting BT development in both the axial and peripheral areas (Figure 5B,C) [68,69]. Appealingly, the first two mutations mentioned accounted for more than 60% of frequencies and were seemingly related to colorectal cancer (codon 146) and numerous cancers (codon 12) [67,70,71].

**Figure 5 cancers-15-04107-f005:**
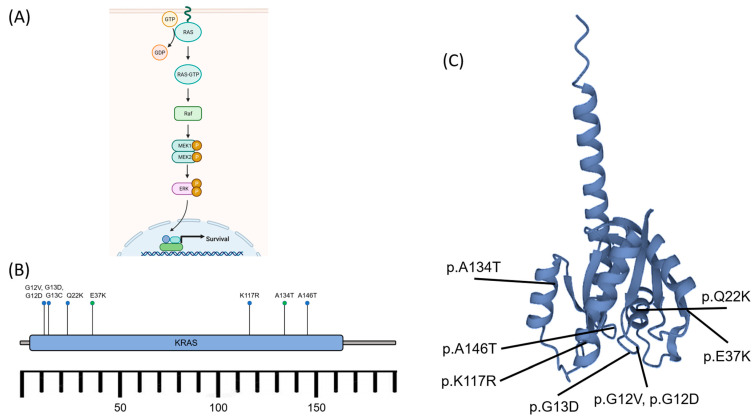
(**A**) Schematic diagram of the RAS activation to induce the MAPK/ERK signalling pathway. The signalling cascade of the pathway leads to nuclear transcription and cell survival. (**B**) The KRAS gene, located on the short arm of chromosome 12, encodes the protein K-ras consisting of ~188–189 amino acids. The activating mutations (blue dots) such as p.A146T, p.G12D/V, p.G13C/D, p.K117R, and p.Q22K affected Brown Tumour development. Some other known mutations (green dots), p.A134T and p.E37K, exhibited no specific significance. (**C**) K-ras protein and the mutation sites. GTP: Guanosine triphosphate, GDP: Guanosine diphosphate, Raf: rapidly accelerated fibrosarcoma kinases (related to Ser/Thr kinases), MEK1/2 (MAPK): Mitogen-activated protein kinase 1/2, ERK: extracellular signal-regulated kinase, P: phosphate (after phosphorylation), KRAS: Kirsten rat sarcoma viral oncogene homolog [63,69,70,72].

Other pathogenic activating mutations such as TRPV4 and FGFR1 have recently been reported in giant-cell lesions of the jaws and non-ossifying fibromas of the bones, which are histologically mimics of Brown Tumours [70]. These mutations have been continuously triggering the activation of ERK1/2 in the MAPK pathway, which results in cell survival [69,70].

The involvement of genetic mutation shifted the prior paradigm of Brown Tumours, as non-neoplastic cell growth due to hyperparathyroidism turned into true neoplasms [69,73]. However, the cause of KRAS mutations in BT remains unclear. KRAS mutations have also been discovered under normal conditions and are associated with aging and smoking [74]. Moreover, certain KRAS variants, p.A134T and p.E37K, also exhibited unknown significance [69].

## 4. Diagnosis: The Role of Radiological Imaging and Lesion Biopsy

OFC and Brown Tumours present multifocal osteolytic, reactive, and non-neoplastic lesions but may sometimes be misdiagnosed as malignant lesions and can mimic histopathologic features of giant cell tumours [75]. The appearance of the osteolytic lesion may coincide with a solid aneurysmal bone cyst, and giant cell reparative granulomas are included in the differential diagnoses for Brown Tumours [76,77]. Based on radiology findings, OFC can also be mistakenly diagnosed as a multiple scattered osteolytic lesion of metastatic disease, e.g., multiple myeloma, breast cancer, renal cell carcinoma, and thyroid carcinoma [78,79]. The laboratory results, i.e., serum PTH, calcium, and phosphorus concentrations, may play a fundamental role in narrowing down the diagnosis [75,77,78,79,80]. However, the advancement of radiological imaging and a tissue biopsy may be significantly beneficial for establishing the diagnosis of OFC and Brown Tumours.

OFC and Brown Tumours can have significant consequences if left untreated, including bone pain, fractures, and deformities. Therefore, diagnosing these conditions as early as possible is important to initiate the appropriate treatment and prevent complications.

### 4.1. Radiological Imaging

The diagnosis of OFC and Brown Tumour typically begins with a thorough medical history and physical examination. The presence of CKD and hyperparathyroidism should be evaluated, as these conditions are associated with an increased risk of developing these bone lesions [81,82]. Imaging studies are also an important part of the diagnostic process. X-rays and computed tomography (CT) can help identify the presence of bone lesions and cysts. OFC is seen on X-rays or CT scans as lytic or multi-lobular cystic alterations (Figure 1 and Figure 2). On a CT scan, numerous bony lesions that are Brown Tumours may be incorrectly identified as metastatic cancer, bone cysts, osteosarcoma, and particularly giant-cell tumours of the bone [25,81,82]. The concurrent measurement of the parathyroid hormone will help to distinguish between primary hyperparathyroidism and cancer [81]. However, these imaging modalities may not be sensitive enough to detect early bone changes.

Magnetic resonance imaging (MRI) is a more sensitive imaging modality for the detection of bone lesions in patients with CKD. The MRI of patients with OFC shows multiple bony osteolytic lesions and an enhanced mass-like change in the affected bone [16]. The MRI can provide detailed information on the extent and severity of bone changes, and it can also help to differentiate between OFC and Brown Tumours.

Parathyroid scintigraphy, a nuclear imaging method utilizing emitted gamma rays, involves a number of different radiotracers and protocols [83]. Single-photon emission computed tomography (SPECT) tracers include ^99^mTc-sestamibi or ^99^mTc-tetrofosmin for parathyroid localization, while PET tracers include ^11^C-methionine (11C-MET), ^18^F-fluorodeoxyglucose (^18^F-FDG), ^18^F-fluorocholine (^18^F-FCH), or ^18^F-sodium fluoride (^18^F-NaF). The ^99^mTc-Sestamibi SPECT and ^11^C-MET PET/CT are being highlighted as the most common tracers used for the detection of parathyroid adenomas [83,84,85].

The intraoperative ^99^mTc-Sestamibi scanning may improve the curative effect of guided parathyroidectomy, although reports show ^99^mTc-Sestamibi to have a low sensitivity of about 40–50% in detecting hyperplastic glands [86,87]. Using carbon as a labelled tracer is very demanding due to its short half-life (20 min), compared to fluorine (109.7 min) [83,84,85,86,87,88]. A meta-analysis of ^11^C-MET PET exhibited an estimated sensitivity of 60–78%, suggesting ^18^F-FCH-PET with a greater accuracy of 77–89% [85,89]. Increased cell proliferation and metabolism in the adenoma or hyperplasia leads to increased choline uptake of ^18^F-FCH PET/CT, either in primary or secondary hyperparathyroidism [90,91]. The extended signal uptake on ^18^F-FDG PET/CT seen within Brown Tumours may be indistinguishable from bone metastases [84]. Although lacking substantial data, the use of ^18^F-NaF PET/CT in hyperparathyroidism and OFC is promising due to its ability to show either increased osteoblast activity or high vascularization in the lesions [92,93,94,95].

### 4.2. Tissue Biopsy

A core needle biopsy is the gold standard for the diagnosis of fibrosa cystica osteitis and Brown Tumours [96]. The fine needle biopsy (aspiration cytology) may suggest OFC appears as an ossifying fibroma displaying spindled neoplastic cells producing trabecular bone and cementoid areas [97,98,99]. Brown Tumours are highly vascularized lesions, histologically characterized by increased—often clustered—osteoclastic activity, haemorrhage, fibrosis, and reactive woven bone formation [97,98]. In a Brown Tumour, the degradation of red blood cells will appear as brown pigment as the result of an accumulation of hemosiderin [97,98]. The histological differential diagnosis of Brown Tumour includes giant cell Tumour of bone, giant cell reparative granuloma, or solid aneurysmal bone cyst. Clinical details include blood chemistry, radiological examination, and eventually fluorescence in situ hybridization (FISH) for USP6 (ubiquitin-specific protease) rearrangements or immunohistochemistry for H3F3AG34 mutated protein [100].

In summary, the diagnosis of OFC and Brown Tumours in CKD requires a combination of medical history, physical examination, imaging studies, and biopsy. This approach can help to accurately diagnose these conditions and initiate the appropriate treatment to prevent complications.

## 5. Treatment: Beyond Removal of the Tumour

### 5.1. Pharmacological Prevention

There are potential pharmacological treatments for OFC and Brown Tumours in CKD, ranging from Biphosphonate, Calcimimetics, and Vitamin D supplementation to Denosumab.

Bisphosphonates are a class of medications that are commonly used to treat osteoporosis and other bone conditions. They work by inhibiting the activity of osteoclasts, which are the cells responsible for the breakdown and removal of bone tissue [101,102]. In CKD, bisphosphonates can help prevent excessive osteoclast activation and the subsequent development of OFC and Brown Tumours [101,102].

Calcimimetics are a relatively new class of medications that are designed to regulate calcium and phosphorus levels in the body [103]. They work by activating the CaSR in the parathyroid glands, which can help reduce the production of PTH and restore the balance of calcium and phosphorus metabolism [104]. In CKD, calcimimetics can help prevent the development of hyperparathyroidism and the subsequent development of OFC and Brown Tumours [104,105].

Vitamin D is an essential nutrient that is involved in the regulation of calcium and phosphorus metabolism. In CKD, vitamin D deficiency can contribute to the development of hyperparathyroidism and the subsequent development of OFC and Brown Tumour [106,107,108]. Supplementation with vitamin D can help restore normal levels of calcium and phosphorous.

Denosumab is a monoclonal antibody specifically designed to target RANKL, which is a protein that plays a key role in osteoclast activation [109]. By blocking the action of RANKL, denosumab can help reduce osteoclast activity and prevent the development of OFC and Brown Tumours. However, there have not been any studies specifically looking at how well it prevents fractures [110]. There have also been reports of denosumab-induced hypocalcemia, which disproportionately affects those with ESKD [110]. Lower baseline blood calcium and 25-hydroxyvitamin D levels, as well as both low and high bone turnover, are risk factors for hypocalcemia with denosumab usage in CKD [110]. If considering denosumab, it is crucial to choose the “appropriate patient,” supplement with calcium and vitamin D, modify calcium dialysate, and undergo continuous clinical monitoring.

In summary, bisphosphonates, calcimimetics, vitamin D, and denosumab are potential pharmacological treatments for OFC and Brown Tumours in CKD. These medications can help prevent excessive osteoclast activation and restore the balance of calcium and phosphorus metabolism, which can prevent the development of these conditions. More clinical research is needed to determine the effectiveness and safety of these treatments for patients with CKD.

### 5.2. Surgical Treatment

A meta-analysis of the treatment options for OFC and Brown Tumours involves the removal of the parathyroid gland, cyst drainage, and bone grafting [14].

Monoclonal proliferation with nodular hyperplasia and reduced expression of vitamin D and CaSR are likely to present when parathyroid hormone levels continue above 800 pg/mL for more than 6 months despite extensive medical interventions [111]. Therefore, a surgical parathyroidectomy should be taken into consideration, especially if other conditions like persistent hyperphosphatemia or hypercalcemia, vascular or tissue calcification, including calciphylaxis, and/or progressive osteodystrophy are present [111].

Parathyroidectomy is a surgical procedure involving the removal of one or more parathyroid glands [112]. By removing the source of excess PTH production, parathyroidectomy can help restore the balance of calcium and phosphorus metabolism and prevent the development of these conditions [112]. Furthermore, patients on dialysis who get a parathyroidectomy have a 15–57% higher chance of surviving, as well as improved hypercalcemia, hyperphosphatemia, tissue calcification, bone mineral density, and health-related quality of life [111].

In patients with OFC, cystic lesions may form in the affected bones. These cysts can cause pain and discomfort and can also increase the risk of fractures [20]. Cyst drainage is a surgical procedure that involves the removal of fluid from these cysts to alleviate symptoms and prevent complications [20].

In patients with advanced OFC or Brown Tumours, significant amounts of bone tissue may be lost. This can lead to deformities and a decreased ability to bear weight. Bone grafting is a surgical procedure involving bone tissue transplantation from a donor site to the affected area [20]. This can help restore the structural integrity of the bone and improve function [20].

## 6. Conclusions

OFC and Brown Tumours are bone lesions found in CKP patients. The risk factors for developing these two conditions include age, sex, a medication that disrupts calcium metabolism, and a vitamin D deficiency. Secondary hyperparathyroidism is the main mechanism leading to the imbalance of calcium and phosphorous levels, which may cause osteoclast activation. However, other mechanisms, such as RAAS hyperactivity and chronic inflammation, may also contribute to the formation of OFC and Brown Tumours in CKD. Given those mechanisms, pharmacologic treatments such as bisphosphonate, calcimimetics, vitamin D supplementation, and denosumab could be utilized to attenuate hyperparathyroidism, restore the calcium level, and prevent the occurrence of OFC. The rare prevalence of Brown Tumours causes a lack of understanding about the manifestation and treatment of this disease. The Pathophysiology of several mechanisms involved in Brown Tumours in CKD patients is also not well understood. However, considering the life-quality impact of this disease in CKD patients, it is important for nephrologists and medical practitioners working with dialysis patients to be aware of various options for diagnosis and treatment. Furthermore, more research is needed in terms of the comparative efficacy and safety of the treatment options in various populations of CKD patients.

## Figures and Tables

**Figure 1 cancers-15-04107-f001:**
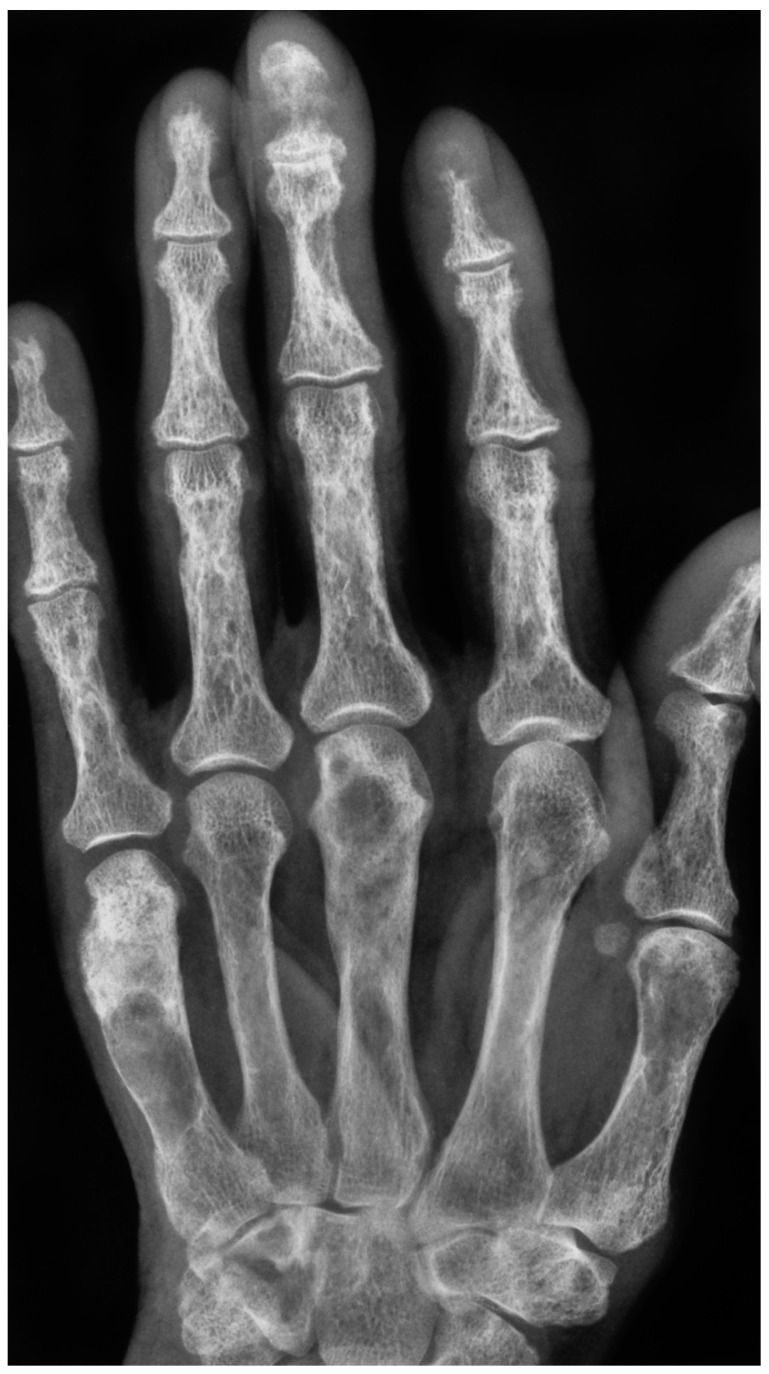
X-ray of a 66-year-old female showing multiple radiolucencies in the metacarpals and phalanges (Brown Tumours) and subperiosteal erosions, with a lace-like appearance of the cortex (Osteitis fibrosa cystica), quite typical for hyperparathyroidism.

**Figure 2 cancers-15-04107-f002:**
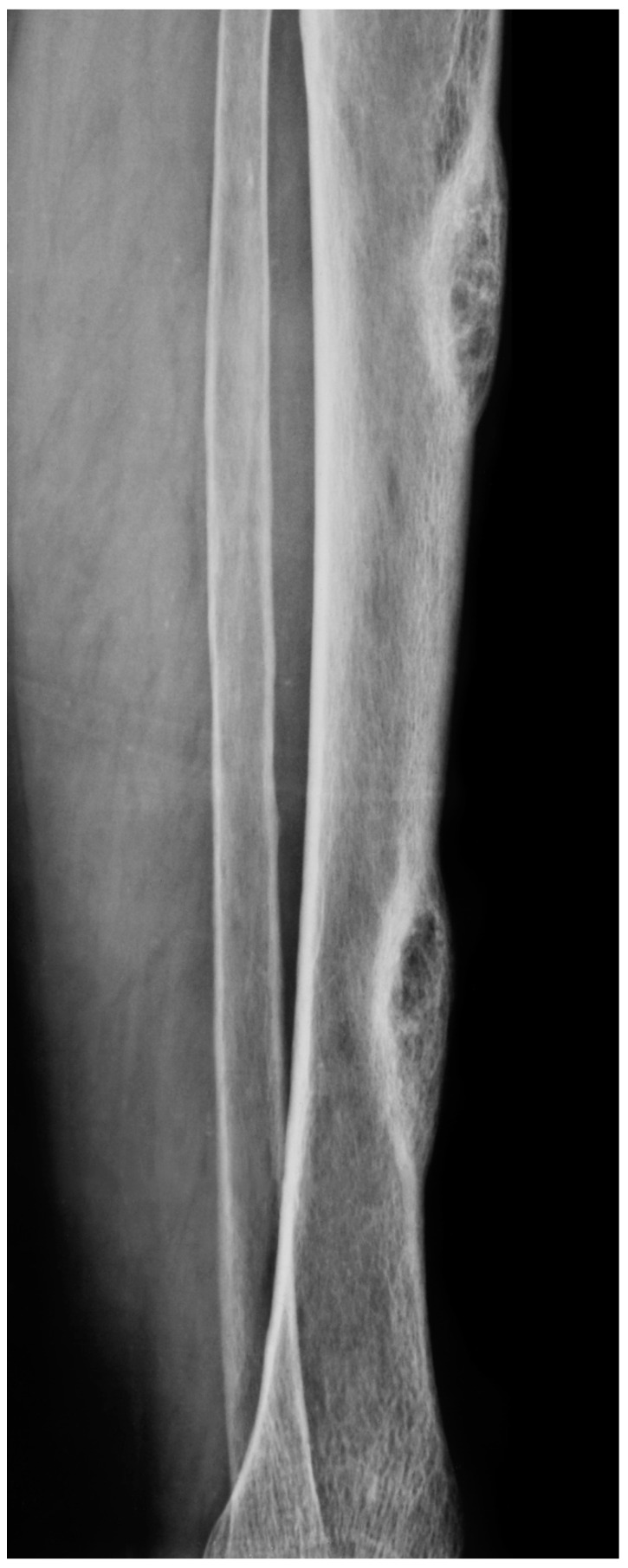
A. X-ray of a 70-year-old female showing two diaphyseal well-circumscribed cortical radiolucencies eroding the cortical bone and with a thick radiopaque inner margin, representing Brown Tumours.

**Figure 3 cancers-15-04107-f003:**
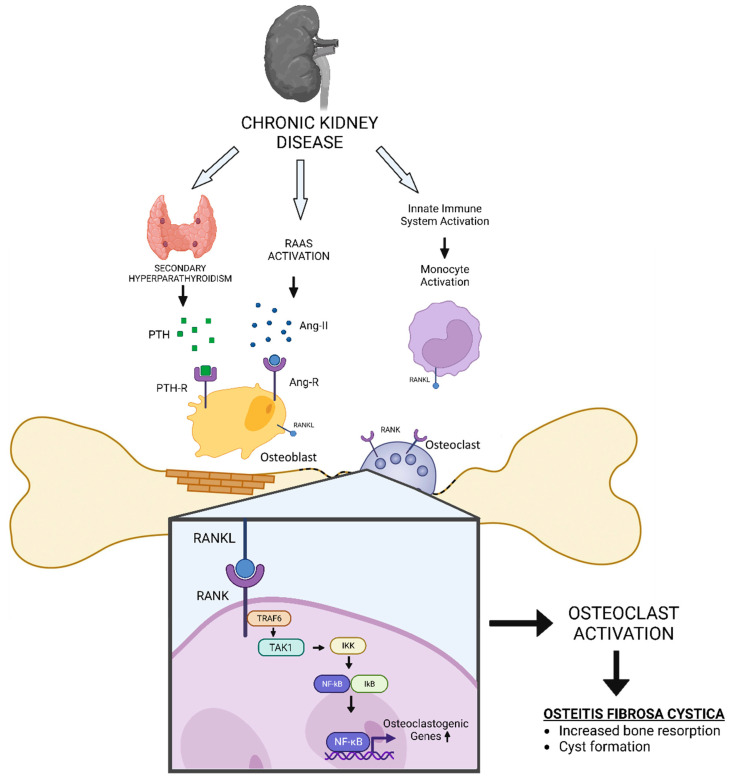
Schematic diagram of how CKD could induce Osteitis Fibrosa Cystica (OFC).

**Figure 4 cancers-15-04107-f004:**
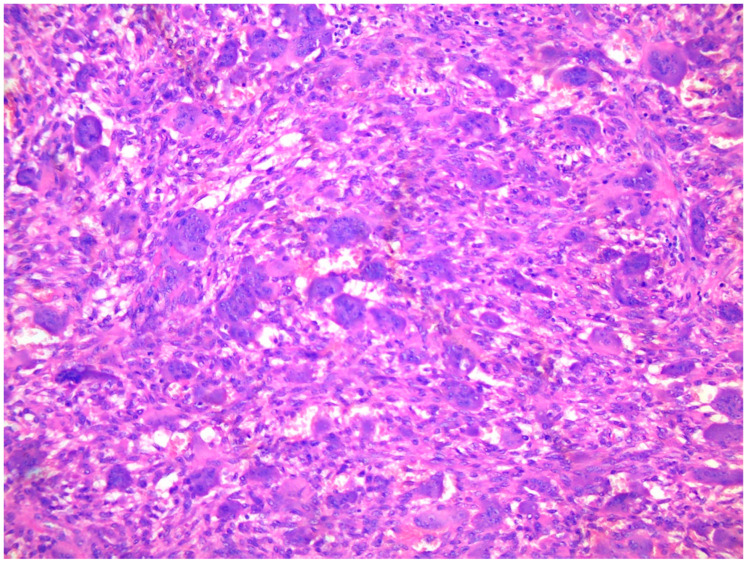
Histology of a typical Brown Tumour-associated with hyperparathyroidism showing scattered multinucleated osteoclasts in a slightly storiform arranged background of spindled and rounded pre-osteoclasts.

## Data Availability

Not applicable.

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
