# Peer review of "Brown Tumour in Chronic Kidney Disease: Revisiting an Old Disease with a New Perspective"

_cancers, 2023, doi:10.3390/cancers15164107_

Round 1

Reviewer 1 Report

You selected an interesting theme that reminds us of an old but still present foe in the progression of CKD, which should not be overlooked during the assessment of patients diagnosed with CKD-MBD. There are some few suggestions related to your work:

Lines 59-60: "The most common cause of chronic kidney disease is diabetes mellitus, followed by hypertension and glomerulonephritis" - even if it is a true statement, it seems just inserted in the text, without any relevance for the present subject. If you would like to keep this phrase, better to highlight the correlation between the primary renal diseases you mentioned and CKD-MBD onset (which of these conditions have a higher risk of presenting secondary hyperparathyroidism etc.), and consequently the risk of developing OFC.

Lines 112-115 ("Previous studies have suggested .... ") - a similar content you have already mentioned in lines 71-72 and lines 80-82... better to emphasise that this statement was already mentioned, such as "As previously mentioned ..." or "Considering that ..." or just delete it and just present the mechanisms (beyond hyperthyroidism) that can have an important role in OFC development.

Please link Figures 2, 3 and 4 with the written text (as you did for Figure 1).

Please explain FISH and USP6 abbreviations inside the text, such as "fluorescence in situ hybridisation (FISH) for USP6 (ubiquitin-specific protease) gene".

CaSR - you have already explained this abbreviation in line 124, so in lines 259-260 you can use directly the abbreviation.

The same for RANKL - please explain the abbreviation in the manuscript only one time when it was initially used. Please check again your whole manuscript for repeated abbreviations, and verify that the explanations were mentioned only one time when they were firstly inserted in the text.

Author Response

I am writing in response to the review of our manuscript entitled "Brown Tumour in chronic kidney disease: revisiting an old disease with a new perspective," which we submitted to Cancers. We sincerely appreciate the time and effort the editorial team has invested in evaluating our work. In this cover letter, we address each of the comments and describe the changes we have made to the manuscript accordingly.

  1. Relevance of References:

We have reviewed all references cited in the manuscript to ensure their relevance to the contents of the manuscript and added a few recent ones after an extensive literature search; We are pretty confident that we are up to date and complete here now.

  1. Highlighted Revisions:

To facilitate the reviewing process, we have carefully highlighted all the revisions made in the manuscript, making it easier for the editors and reviewers to identify and evaluate the modifications.

  1. Cover Letter for Detailed Revisions:

In compliance with the editor's request, we have prepared this cover letter to provide a point-by-point explanation of the revisions made to the manuscript and our responses to the comments.

  1. Simple Summary:

We have added a simple summary to the manuscript. This provides a clear and accessible overview of the manuscript to enhance its readability and impact.

  1. Word Count Extension:

To meet the journal's requirements for a review manuscript, we have extended the words in the main text to a minimum of 4000 words.

  1. Author Contribution:

We have included an "Author Contributions" section at the back matter of the manuscript, as requested. This section outlines the specific contributions of each author to the research, acknowledging their individual roles in the conception, design, data collection, analysis, and writing of the manuscript.

  1. Citation of Figures 1-5:

To ensure completeness and accuracy in the manuscript, we have added proper citations for Figures 1 to 5 in the main text.

  1. Elaborate on the relationship between CDK and Brown tumour formation.

We more extensively elaborated on the mechanisms and added an explanation of this relationship in an additional paragraph.

  1. English

We edited some English grammar, corrected a typo in one of the author's names, and had the manuscript proofread by a native speaker.

  1. Abbreviations

They were introduced at first mention.

We believe that the revised version of the manuscript now meets the standards set by Cancers and is ready for re-evaluation. We thank the editor for the insightful comments. Please find attached the revised manuscript along with the highlighted changes. Thank you for your time and consideration.

Reviewer 2 Report

The brown tumor is an extreme form of osteitis fibrosa cystica, representing a serious complication of the advanced primary or secondary hyperparathyroidism. The link between Brown Tumor and chronic kidney disease is novel. In this review, Santoso et al elaborated the mechanism of several brown tumour that related with CKD, such as the hyperparathyroidism, RAAS activation, inflammatory factors, and so on. This is a well done, in dept study. There are only a few points that should be addressed.

1.     I don't know much about Brown Tumor, but from this review, I didn't see direct data that CKD patients have Brown Tumor. I hope the author can provide more patient data.

2.     The author selected some references that are not related to kidney disease to support his point of view. For instance, in line 117, the author elaborates on Secondary hyperparathyroidism that classically occurred during CKD is central in 117 the formation of Brown Tumor, but there is no literature reference in the whole paragraph. Although the author gives an example below that CKD can cause secondary HPT, and secondary HPT is related to Brown Tumour, this does not prove that Secondary hyperparathyroidism that classically occurred during CKD is central in 117 the formation of Brown Tumour. Authors need to provide direct evidence of literature.

3.     I suggest that the author elaborate on the limitation of brown tumor and CKD.

4.     In my opinion, the references of the author are somewhat outdated. I recommend the author to increase the proportion of papers from the past five years.

Author Response

I am writing in response to the review of our manuscript entitled "Brown Tumour in chronic kidney disease: revisiting an old disease with a new perspective," which we submitted to Cancers. We sincerely appreciate the time and effort the editorial team has invested in evaluating our work. In this cover letter, we address each of the comments and describe the changes we have made to the manuscript accordingly.

  1. Relevance of References:

We have reviewed all references cited in the manuscript to ensure their relevance to the contents of the manuscript and added a few recent ones after an extensive literature search; We are pretty confident that we are up to date and complete here now.

  1. Highlighted Revisions:

To facilitate the reviewing process, we have carefully highlighted all the revisions made in the manuscript, making it easier for the editors and reviewers to identify and evaluate the modifications.

  1. Cover Letter for Detailed Revisions:

In compliance with the editor's request, we have prepared this cover letter to provide a point-by-point explanation of the revisions made to the manuscript and our responses to the comments.

  1. Simple Summary:

We have added a simple summary to the manuscript. This provides a clear and accessible overview of the manuscript to enhance its readability and impact.

  1. Word Count Extension:

To meet the journal's requirements for a review manuscript, we have extended the words in the main text to a minimum of 4000 words.

  1. Author Contribution:

We have included an "Author Contributions" section at the back matter of the manuscript, as requested. This section outlines the specific contributions of each author to the research, acknowledging their individual roles in the conception, design, data collection, analysis, and writing of the manuscript.

  1. Citation of Figures 1-5:

To ensure completeness and accuracy in the manuscript, we have added proper citations for Figures 1 to 5 in the main text.

  1. Elaborate on the relationship between CDK and Brown tumour formation.

We more extensively elaborated on the mechanisms and added an explanation of this relationship in an additional paragraph.

  1. English

We edited some English grammar, corrected a typo in one of the author's names, and had the manuscript proofread by a native speaker.

  1. Abbreviations

They were introduced at first mention.

We believe that the revised version of the manuscript now meets the standards set by Cancers and is ready for re-evaluation. We thank the editor for the insightful comments. Please find attached the revised manuscript along with the highlighted changes. Thank you for your time and consideration.

Sincerely,